# The Synaptic and Intrinsic Cellular Mechanisms of Persistent Firing in Neurogliaform Cells

**DOI:** 10.3390/biom15111603

**Published:** 2025-11-15

**Authors:** Shiyuan Chen, Xiaoshan Chen, Jianwen Zhou, Jinzhao Wang, Kaiyuan Li, Wenyuan Xie, Cheng Long, Gangyi Wu

**Affiliations:** 1School of Life Sciences, South China Normal University, Guangzhou 510631, China; 2Greater Bay Area Institute of Precision Medicine (Guangzhou), School of Life Sciences, Fudan University, Guangzhou 511458, China; 3Guangdong Engineering Research Center of Precision Detection and Modulation of Human Microbiome, South China Normal University, Guangzhou 510631, China

**Keywords:** neurogliaform cells, GABAergic neuron, persistent firing, T-type calcium channel, hippocampus, afterdepolarization

## Abstract

While persistent firing in glutamatergic neurons has been well-characterized, the intrinsic and synaptic mechanisms driving this phenomenon in neurogliaform cells (NGFCs), a subtype of GABAergic interneurons, remain unclear. This study investigates the mechanisms underlying persistent firing in hippocampal NGFCs. Whole-cell current-clamp recordings were performed on acute brain slices from C57BL/6J mice to examine the electrophysiological properties of NGFCs in the hippocampal stratum lacunosum-moleculare (SLM). Pharmacological interventions, including T-type calcium channel blocker ML218 and 5-hydroxytryptamine (5-HT) receptor antagonist olanzapine, were used to dissect the mechanisms of persistent firing. Biocytin labeling and confocal microscopy were employed to confirm neuronal morphology and location. The study revealed that persistent firing in NGFCs is induced by a long-lasting delayed afterdepolarization (L-ADP), which depends on T-type calcium channels (intrinsic mechanism) and is modulated by 5-HT receptors (synaptic mechanism). Persistent firing was observed in 62.96% of SLM neurons and was abolished by ML218 or olanzapine. The findings bridge a gap in understanding how inhibitory interneurons contribute to memory processes. The dual-mechanism framework (T-type channels and 5-HT receptors) aligns with prior work on glutamatergic systems but highlights unique features of GABAergic persistent firing. These insights advance the understanding of inhibitory circuit dynamics and their potential role in cognitive functions, paving the way for further research into interneuron-specific memory encoding.

## 1. Introduction

The brain, a complex multi-scale system, is essential for cognitive functions such as decision-making, learning, memory, and motor control. Neurons, the fundamental units of neural circuits, are classified into subtypes based on their morphology, physiology, molecular signatures, or neurotransmitter profiles [1]. When categorized by neurotransmitter systems, neurons are generally classified into glutamatergic, GABAergic, dopaminergic, serotonergic, and cholinergic subtypes [2]. Among these, glutamatergic neurons constitute approximately 40% of all neurons in the brain [3], with higher proportions in the cortex (80%) [4] and hippocampus (90%) [5]. Despite their lower prevalence compared to glutamatergic neurons, GABAergic interneurons exhibit remarkable diversity and play an essential role in neuronal circuits of the cortex and hippocampus [6,7]. The balance between excitatory glutamatergic neurons and inhibitory GABAergic neurons is crucial for normal brain functions, including learning and memory [8,9].

Neurons exhibit a variety of firing patterns that are believed to be relevant to their function. Persistent firing, a particular firing pattern, has been observed in both glutamatergic and GABAergic neuronal systems. In the glutamatergic system, persistent firing is commonly induced by a cholinergic agonist carbachol (CCh) and has been extensively studied in brain regions such as the anterior cingulate cortex (ACC), medial entorhinal cortex, hippocampus, postsubiculum, and amygdala [10,11,12,13]. This type of firing depends on metabotropic receptor 5 (mGluR5) and requires a calcium-dependent plateau potential [14]. In contrast, within the GABAergic system, neurogliaform cells (NGFCs)—a specific subtype of interneurons expressing 5-hydroxytryptamine (5-HT)3a receptors—exhibit delayed persistent firing in response to depolarization stimuli, independent of exogenous agonist or neurotransmitters [15,16]. Notably, this retroaxonal barrage firing, which is a form of burst firing where action potentials (APs) propagate retrogradely from the axonal terminals toward the soma, is resistant to blockade of N-methyl-D-aspartate (NMDA), alpha-amino-3-hydroxy-5-methyl-4-isoxazole propionic acid (AMPA), gamma-aminobutyric acid (GABA)_A_, and GABA_B_ receptors, highlighting a distinct mechanism [17,18,19]. However, the molecular and cellular underpinnings of GABAergic persistent firing remain poorly defined.

Here, we investigate the mechanisms underlying persistent firing in hippocampal NGFCs. We demonstrate that long-lasting delayed afterdepolarization (L-ADP), also termed a plateau potential, induces persistent firing via T-type calcium channels (intrinsic mechanism), while 5-HT receptor activation (synaptic mechanism) modulates this process. Our findings not only elucidate the interplay between intrinsic and synaptic mechanisms in inhibitory circuits but also suggest a potential role for NGFCs in short-term memory through transient information storage.

## 2. Materials and Methods

### 2.1. Animals

Two-month-old C57BL/6J mice used in this study were obtained from the Model Animal Research Center of Nanjing University (Nanjing, China). Both female (50%) and male (50%) mice were used in this study. Animals were given ad libitum access to food and water, and were housed on a 12/12 h light/dark cycle (lights on at 8 A.M.) at 23–25 °C and 50–60% humidity.

### 2.2. Acute Brain Slice Preparation

Using a vibratome (Leica VT1000S, Wetzlar, Germany), brain slices (~330 μm thick) of 2-to-3-month-old C57BL/6J mice were cut coronally in ice-cold cutting solution from the dorsal hippocampus and prefrontal cortex. The cutting solution contained (in mM) 92 N-methyl-D-glucamine (NMDG), 93 HCl, 2.5 KCl, 1.25 NaH_2_PO_4_, 30 NaHCO_3_, 20 4-(2-Hydroxyethyl)piperazine-1-ethanesulfonic acid (HEPES), 25 glucose, 2 thiourea, 5 (+)-sodium L-ascorbate, 3 sodium-pyruvate, 0.5 CaCl_2_⋅2H_2_O, and 10 MgSO_4_⋅7H_2_O. After rewarming for 10 min, sectioned slices were transferred to the holding solution containing (in mM) 92 NaCl, 2.5 KCl, 1.25 NaH_2_PO_4_, 30 NaHCO_3_, 20 HEPES, 25 glucose, 2 thiourea, 5 (+)-sodium L-ascorbate, 3 sodium pyruvate, 2 CaCl_2_⋅2H_2_O, and 2 MgSO_4_⋅7H_2_O, and recovered for at least 1 h at room temperature (RT). Artificial cerebrospinal fluid (aCSF) contained (in mM) 124 NaCl, 2.5 KCl, 1.25 NaH_2_PO_4_, 24 NaHCO_3_, 12.5 glucose, 5 HEPES, 2 CaCl_2_⋅2H_2_O, and 2 MgSO_4_⋅7H_2_O, and was used for whole-cell current- or voltage-clamp recording. The final pH of the above solutions was adjusted to 7.3–7.4, osmotic pressure was regulated between 280 and 290 mosm/L, and saturation with 95% O_2_/5% CO_2_ was maintained throughout the experiments [20].

### 2.3. Whole-Cell Current-Clamp Recording

To assess the intrinsic membrane and AP properties of NGFCs, recordings were performed at 32–34 °C with electrodes (5–6 MΩ resistance) pulled from borosilicate glass filled with an intracellular solution of the following composition (in mM): 110 K-gluconate, 40 HEPES, 10 ethylene glycol tetraacetic acid (EGTA), 2 Na_2_-ATP, 2 Mg-ATP and 0.3 GTP (pH 7.35; 280–290 mOsm). Data were collected using a pClamp 10.4 device and were analyzed with Clampfit 10.4 software (Molecular Devices, Palo Alto, CA, USA), sampled at 10 kHz, and filtered at 1 kHz. For recordings in the hippocampus and cortex, the voltage responses were measured at steady state. We applied hyperpolarizing and depolarizing current steps of increasing amplitude (50 pA increase in step size per sweep, 1 s duration, one sweep every 1 s) to examine the firing pattern of recorded cells. Two protocols were designed to study persistent firing, one with increasing step size (50 pA increase in step size per sweep, 1 s step duration, 3 s intersweep interval), and a second with increasing stimulation time (4 s increase per sweep, 5 steps in total, 500 pA injection current). When persistent firing was triggered multiple times in the same cell, a period of 2–3 min was typically allowed after the cessation of one round of persistent firing before the next round of stimulation [21].

### 2.4. Pharmacology

In certain experiments (see Results section for details), the recording solution was supplemented with one or a combination of the following drugs: 20 μΜ 2-methyl-6-(phenylethynyl) pyridine (MPEP, mGlu5 receptor blocker, Tocris, Cat. No. 1212), 10 μΜ LY367385 (mGlu1 receptor blocker, Tocris, Cat. No. 1237), 100 μΜ 2-amino-5-phosphonovalerate (APV, blocker of NMDA ionic glutamate receptor, Sigma, A5282) and 80 μΜ 6-cyano-7-nitroquinoxaline-2,3-dione (CNQX, blocker of alpha-amino-3-hydroxy-5-methylisoxazolepropionic acid (AMPA) ionic glutamate receptors, Sigma, C127), 10 μΜ picrotoxin (GABA receptor blocker, Tocris, Cat. No. 1128), 20 μΜ atropine (cholinergic receptor blocker, MCE, atropine), 20 μΜ domperidone (dopamine D2 receptor blocker, Selleck, S2461), 20 μΜ ML218 (T-type Ca^2+^ channel inhibitor, Tocris, Cat. No. 4507), 20 or 40 μΜ olanzapine (broad-spectrum 5-HT receptor blocker and dopamine D2 receptor blocker, Selleck, S2493). Where indicated, drugs were washed out with recording solution.

All pharmacological experiments were performed on slices from at least 3 animals, with the number of cells and animals specified in the figure legends.

### 2.5. Biocytin Labeling, Image Collection and Analysis

To check neuron morphology after electrophysiological recordings, in some cases, 0.2% biocytin (B4261, Sigma, Ronkonkoma, NY, USA) was added to the intracellular solution, allowing the visualization of patched cells. Slices containing biocytin-filled NGFC interneurons were fixed in 4% paraformaldehyde (PFA) overnight at 4 °C. Then the slices were washed three times (5 min each time) with phosphate-buffered saline (PBS), and transferred to 1% Triton-X100/PBS overnight at 4 °C. Subsequently, slices were incubated at 4 °C overnight in streptavidin-Alexa 488 (S11223, Invitrogen, Carlsbad, CA, USA) diluted 1:1000 with 0.3% Triton-X100/PBS, then washed three times (5 min each time) with PBS and mounted on glass slides. Images were collected with a confocal microscope (LSM-800, Carl Zeiss AG, Jena, Germany) using 20× (2 μm Z-step) and 40× oil immersion (0.2 μm Z-step) objectives at 1024 × 1024 pixels by sequential scanning and then processed using ZEN (Zeiss, Germany) and ImageJ (Fiji) software (NIH, Bethesda, MD, USA) [22].

### 2.6. Statistical Analysis

All data are expressed as the mean ± standard error of the mean (SEM) and were statistically evaluated by Student’s *t*-test, ANOVA (one-way, two-way, two-way repeated measures) with Tukey’s or Sidak’s post hoc multiple comparison test or two-sample Kolmogorov–Smirnov test using GraphPad Prism 8. *p* < 0.05 was considered significant (* *p* < 0.05, ** *p* < 0.01, *** *p* < 0.001). Mean ± SEM values, sample size, *p*-values and statistical methods are defined in the respective results and figure legends.

## 3. Results

### 3.1. Persistent Firing in Prefrontal Cortex

Using whole-cell current-clamp recording, we observed that a subset of recorded cells exhibited persistent firing when the holding potential was switched from 0 mV to −60 mV in prefrontal cortex (PFC) layer 5 brain slices obtained from C57BL/6J mice (Figure 1A). This potential shift also elicited a long-lasting inward current (Figure 1A, right). Biocytin labeling confirmed the recording location and neuronal morphology in the PFC (Figure 1B). To rule out incomplete voltage clamping, we injected depolarizing currents (ranging from 1 to 13 s in duration, with 4 s increments) in current-clamp mode, which similarly elicited persistent firing (Figure 1C). The long-lasting inward current corresponded to a delayed afterdepolarization (L-ADP), also referred to as a plateau potential, in current-clamp recordings. Notably, persistent firing persisted even after blocking ionotropic glutamate and GABA receptors with 100 μM APV, 80 μM CNQX, and 10 μM picrotoxin (Figure 1C). Persistent firing was observed in 14.29% of PFC neurons (7 out of 49 cells from 6 mice; Figure 1D). In two cells exhibiting persistent firing, hyperpolarizing current injections during the firing period revealed small spikelets (Figure 1E–G). These findings suggest that persistent firing in the PFC may arise from L-ADP or persistent small spikelets.

### 3.2. Persistent Firing in Hippocampal SLM NGFCs

Persistent firing, resistant to glutamate and GABA receptor blockade, has been reported in cortical and hippocampal NGFCs [17]. Given the higher density of NGFCs in the hippocampal stratum lacunosum-moleculare (SLM; ~50%) compared to the cortex [23], we focused our investigation on SLM. The location of recorded cells was confirmed by biocytin labeling (Figure 2A). Non-persistent firing neurons displayed long-lasting afterhyperpolarization potentials (L-AHPs), whereas persistent firing neurons exhibited L-ADPs (Figure 2B,C). Persistent firing was observed in 62.96% of SLM neurons (34 out of 54 cells from 9 mice; Figure 2D). This activity was not an artifact, as it could be reliably and repeatedly triggered (≥3 times; Figure 2E) and induced across a range of stimulation frequencies (10–100 Hz; Figure 2F).

### 3.3. The Intrinsic Properties of Hippocampal SLM Neurons

We compared intrinsic properties between persistent and non-persistent firing neurons. A series of current injections ranging from −150 to 200 pA with 50 pA increments revealed that persistent firing neurons displayed significantly higher firing frequencies than non-persistent firing neurons (Figure 3A,B). No differences were observed in resting membrane potential (RMP), AP threshold, AP amplitude, fast afterhyperpolarization (fAHP) and capacitance (Figure 3C–G). However, persistent firing neurons exhibited broader APs (longer rise and decay times; Figure 3H,I), lower input resistance (Figure 3J), and reduced sag potentials (indicating diminished Ih currents; Figure 3K). These distinctions suggest that persistent and non-persistent firing neurons represent distinct GABAergic subtypes in SLM.

### 3.4. Induction of Persistent Firing in SLM NGFCs by L-ADP

To determine whether persistent firing is induced by L-ADP, we measured the activation threshold for different types of APs in persistently firing neurons, including somatic APs (refer to those directly evoked by depolarizing current injected through the recording electrode), synaptic APs (refer to spontaneous APs driven by synaptic inputs) and persistent APs (refer to those occurring during the persistent firing period after the cessation of the evoked current, Figure 4A). Somatic AP thresholds (where we measured the last AP in the series during persistent firing) were approximately 20 mV higher than those of persistent APs, which is consistent with the initiation of firing in distal axonal regions [16]. The thresholds for persistent APs were approximately 5 mV higher than those for synaptic APs (Figure 4B,C). The firing patterns of somatic APs differed from those of persistent APs, but resembled those of synaptic APs, indicating that synaptic-level mechanisms may participate in persistent firing (Figure 4B,D). We subsequently calculated the Δvoltage, which represents the difference between AP threshold and RMP (Δvoltage = threshold − RMP), for both persistent APs and somatic APs. The L-ADP amplitude was higher than the Δvoltage of persistent APs but lower than the Δvoltage of somatic APs in every recorded persistently firing cell (Figure 4E). Moreover, persistent firing failed to be elicited when the RMP was maintained at values more hyperpolarized than −70 mV, since the membrane potential did not reach the threshold required for persistent firing despite the presence of L-ADP. However, persistent firing could be successfully evoked by the same stimulation protocol when a depolarization current was injected to maintain the membrane potential at −60 mV (n = 3 cells) (Figure 4F). Furthermore, persistent firing was terminated upon application of a 100 ms hyperpolarizing pulse during the ongoing persistent firing period (Figure 4G). Taken together, the above data suggest that L-ADP plays a critical role in the initiation of persistent firing.

### 3.5. The Amplitude of L-ADP in SLM NGFCs Exhibits a Stimulation Intensity-Dependent Relationship

Subsequent to demonstrating the induction of persistent firing by L-ADP, we investigated the potential correlation between L-ADP amplitude and stimulation intensity. Initially, cells were injected with a 1 s duration of depolarization currents ranging from 0 pA to 400 pA in 50 pA steps. L-ADP was either absent or exhibited minimal amplitude at 50 pA stimulation intensity; but the amplitude of L-ADP progressively increased with higher stimulation intensities (Figure 5A). Subsequently, while maintaining a constant depolarization current, we extended the stimulation duration from 1 s to 13 s in 4 s steps. This protocol also induced persistent firing (Figure 5B). Quantitative analysis revealed a strong positive correlation between stimulation intensity and both L-ADP amplitude and persistent-firing spike number (Figure 5C,D). Persistent firing was triggered by membrane depolarization to approximately −10 mV (range: −25 to −5 mV from RMP; Figure 5E).

### 3.6. Synaptic and Intrinsic Cellular Mechanisms Underlying Persistent Firing in SLM NGFCs

Persistent firing in SLM NGFCs was resistant to blockade of ionotropic glutamate or GABA receptors, including 100 μM APV (a glutamate NMDA receptor antagonist), 80 μM CNQX (an AMPA/KA receptor antagonist) or 10 μM picrotoxin (a blocker of GABAA receptors). We also tested 20 μM MPEP and 10 μM LY367385, which are inhibitors of metabolic glutamate receptors, mGluR5 and mGluR1, respectively, but observed no effect. To further explore the mechanisms, we even applied 20 μM atropine (an inhibitor of cholinergic receptors), and 20 μM domperidone (a blocker of dopamine D2 receptors), but neither drug suppressed persistent firing (Figure 6A). Interestingly, we found that olanzapine, a 5-HT receptor blocker, partially blocked persistent firing in SLM NGFCs at 20 μM and fully blocked it at 40 μM (Figure 6B). Indeed, olanzapine not only suppressed persistent firing but also abolished the L-ADP (Figure 6D,E). Despite extensive washing to remove olanzapine, persistent firing failed to recover following inhibition. Our data may, however, provide the evidence for synaptic mechanisms in regulating GABAergic neuron activity in the SLM.

We next tested the effects of L-type and T-type calcium channel blockers on NGFCs. In contrast to the L-type calcium channel blocker nifedipine, which had no effect, the T-type calcium channel blocker ML218 significantly inhibited persistent firing (Figure 6C,F,G). Moreover, the blocking effect of ML218 could be reversed upon washed out. Additionally, since the internal solution contains 10 mM EGTA, it is likely that extracellular calcium may play an important role in the intrinsic cellular mechanisms underlying persistent firing.

Together, our data demonstrate that persistent firing in SLM NGFCs is supported by both synaptic and intrinsic cellular mechanisms.

## 4. Discussion

Persistent firing in NGFCs has garnered significant interest due to its potential role in short-term memory and cognitive processes. While prior studies established the existence of this phenomenon [16,24,25], the mechanisms driving GABAergic persistent firing remained unresolved. However, the mechanisms underlying this phenomenon remain elusive and likely involve a combination of intrinsic cellular mechanisms and synaptic interactions [26]. Our study provides critical insights by demonstrating that L-ADP, mediated by T-type calcium channels (intrinsic mechanism) and modulated by 5-HT receptors (synaptic mechanism), underlies persistent firing in hippocampal NGFCs.

Sheffield and colleagues initially reported persistent firing in SLM interneurons in 2011, noting that it could not be prevented by reducing extracellular calcium concentrations [16]. They later demonstrated that persistent firing could be completely blocked with 100 µM cadmium, while 30 µM nifedipine and calcium-free solutions reduced the number of persistent-firing spikes by 90%, without completely inhibiting the activity [24]. Furthermore, by using 40 µM verapamil, these authors concluded that L-type calcium channel antagonists can inhibit persistent firing. Nifedipine is a specific L-type calcium channel blocker, while verapamil can block both L-type calcium channels [27] and T-type calcium channels [28]. In the current study, we used 20 µM ML218, a specific inhibitor of T-type calcium channels, to block persistent firing in hippocampal SLM interneurons. Thus, we have demonstrated that T-type calcium channels represent the intrinsic cellular mechanism of persistent firing in NGFCs. Both we and Sheffield et al. [24] found that extracellular calcium may play a significant role in persistent firing, because EGTA or BAPTA in piped chelated the intracellular calcium and cannot block persistent firing. But how the calcium influx associated with persistent firing is regulated is currently unclear.

Moreover, another finding of our study was that the kinetics of APs did not strictly follow predictions based on passive membrane properties alone. Specifically, we observed that a lower membrane time constant (τm) did not invariably result in shorter AP rise and decay times. While a reduced τm, resulting from a lower input resistance, should in principle accelerate the kinetics of passive membrane charging, the AP is an active response governed by voltage-gated ion channels. The rise time is primarily a function of voltage-gated sodium (NaV) channel activation kinetics, while the decay time is shaped by NaV channel inactivation and the activation of voltage-gated potassium (KV) channels. Therefore, our results strongly suggest that the differences in AP kinetics we report are dominated by cell-type or compartment-specific variations in the density, subtype composition, or regulatory states of these key ion channels. This highlights that active ionic mechanisms can override the modulating influence of passive cable properties in determining the temporal profile of neuronal spikes.

Our observation that a depolarization stimulation induces an L-ADP, which in turn activates T-type calcium channels leading to persistent firing, is consistent with reports from other laboratories that have observed L-ADP or plateau potentials inducing persistent firing in the glutamatergic system [29]. While the molecular mechanism of L-ADP remains to be elucidated, our data suggest that L-ADP may also play a role in the GABAergic system.

A novel finding of our work is the involvement of 5-HT receptors in L-ADP induction. Specifically, olanzapine, a broad-spectrum 5-HT receptor antagonist, suppressed both L-ADP and persistent firing. Given that NGFCs express 5-HT3a receptors [15], we propose that serotoninergic signaling synergizes with T-type channel activity to amplify depolarization, thereby enabling sustained firing. However, the side effects of olanzapine that reduce AP firing frequency must be considered when interpreting these results [30]. Future studies utilizing subtype-specific 5-HT3a antagonists are necessary to validate this synaptic mechanism.

Notably, our results suggest two distinct modes of persistent firing in NGFCs. The first pattern involves strong somatic depolarization activating the 5-HT3a receptor, leading to the influx of cations and inducing an L-ADP that depends on T-type calcium channels. The second pattern suggests that hundreds of APs may trigger long-lasting persistent dendritic small spikelets, again involving T-type calcium channels [16]. This hypothesis suggests that both intrinsic cellular and synaptic mechanisms can contribute to the persistent firing of NGFCs.

Overall, our study unravels a dual-mechanism framework for GABAergic persistent firing, bridging intrinsic ion channel dynamics and neuromodulatory synaptic mechanism. This discovery advances our understanding of how inhibitory interneurons contribute to memory processes, particularly in the context of short-term memory, where persistent firing may play a crucial role in facilitating the transient storage of information.

## 5. Conclusions

This study elucidates the dual mechanisms underlying persistent firing in hippocampal NGFCs, demonstrating its dependence on intrinsic T-type calcium channels and modulation by 5-HT receptor-mediated synaptic activity. The discovery of long-lasting afterdepolarization (L-ADP) as a trigger for GABAergic persistent firing expands our understanding of inhibitory interneuron contributions to memory processes. These findings bridge a critical gap between glutamatergic and GABAergic persistent firing mechanisms, highlighting the specialized role of NGFCs in neural circuit dynamics. The study provides a foundation for future investigations into interneuron-specific memory encoding and potential therapeutic targets for cognitive disorders.

## Figures and Tables

**Figure 1 biomolecules-15-01603-f001:**
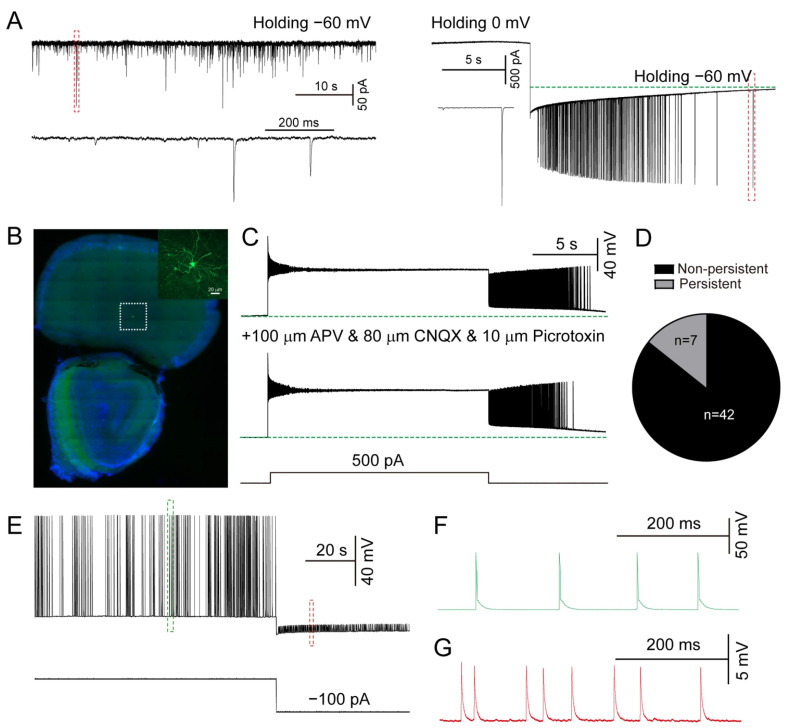
Persistent firing in prefrontal cortex. (**A**) Left, typical traces of spontaneous excitatory postsynaptic currents holding at −60 mV; right, persistent firing was induced when the holding potential was switched from 0 mV to −60 mV in a cortical neuron. The green line represents the baseline potential holding at −60 mV. (**B**) Biocytin staining showing the recording location of PFC and neuronal morphology. (**C**) Persistent firing in cells is resistant to blockade by APV, CNQX and picrotoxin. The green line indicates the voltage recorded in the absence of current injection. (**D**) Quantification of proportion of the two cell types. (**E**) Typical traces of small spikelets recorded during hyperpolarization (approximately −80 mV). (**F**) Persistent firing APs from (**E**). (**G**) Persistent small spikelets from (**E**). (n = 49 neurons from 6 mice, 7 neurons exhibited persistent firing ).

**Figure 2 biomolecules-15-01603-f002:**
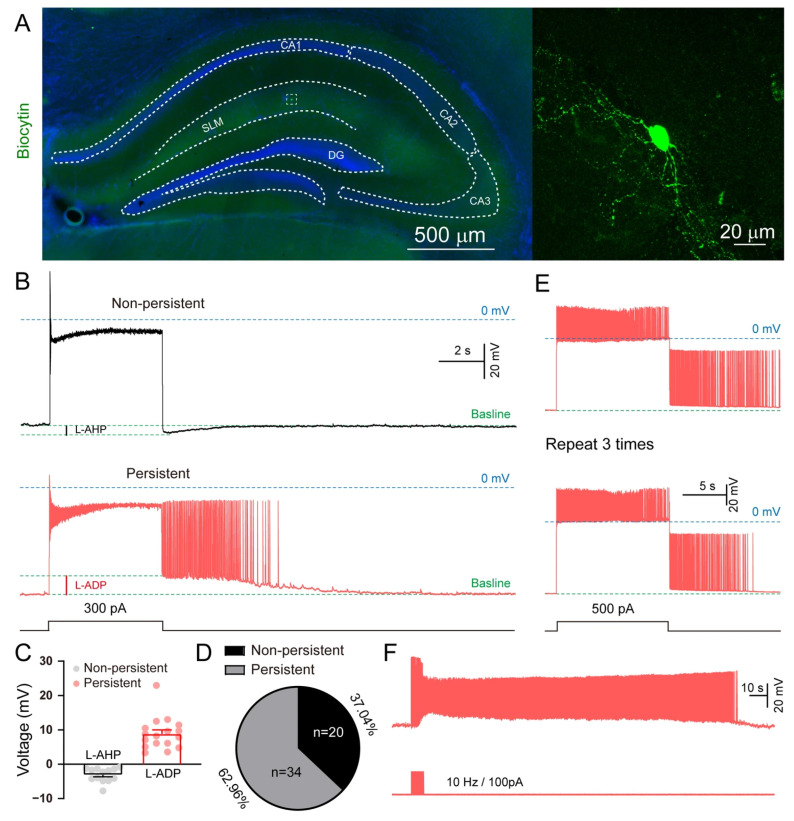
Persistent firing in hippocampal SLM neurons. (**A**) Images of a persistently firing cell labeled with biocytin in SLM. (**B**) Typical traces of non-persistently firing and persistently firing cells. (**C**) Quantification of L-AHP and L-ADP shown in (**B**). (**D**) Quantification of cell population between non-persistent and persistent cells. (**E**) Persistent firing can be repeatedly induced (more than three times) in the same neuron. (**F**) Representative record of persistent firing at a frequency 10 Hz. (n = 54 neurons from 9 mice, 34 neurons exhibited persistent firing).

**Figure 3 biomolecules-15-01603-f003:**
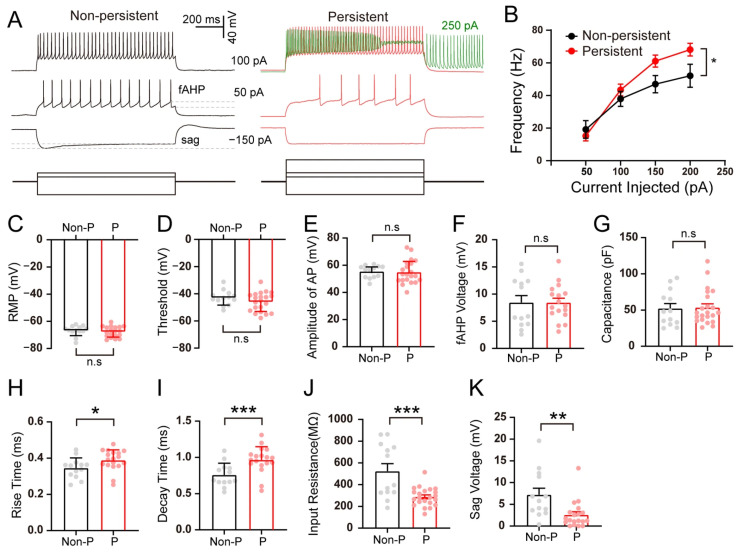
The intrinsic properties of persistently firing and non-persistently firings neurons in SLM. (**A**) Typical traces of firing frequency in response to injected currents (duration 1 s). For the 50 pA trace, the gray dashed lines represent the measurement of the fast afterhyperpolarization (fAHP). For the −150 pA trace, the gray dashed lines represent the measurement of the sag potential. (**B**) Quantification of the current-frequency (I-F) relationship shown in (**A**). All traces in (**A**,**B**) were elicited by short-duration (1 s) current injections. (**C**–**G**) Quantification of RMP, AP threshold, AP amplitude, fAHP and membrane capacitance in (**A**). (**H**–**K**) Quantification of AP rise time, AP decay time, input resistance and sag potential in (**A**). All the APs were somatic Aps; refer to Figure 4. (n = 35 neurons from 6 mice, 22 neurons show persistent firing, ns means no significant difference, * *p* < 0.05, ** *p* < 0.01, *** *p* < 0.001).

**Figure 4 biomolecules-15-01603-f004:**
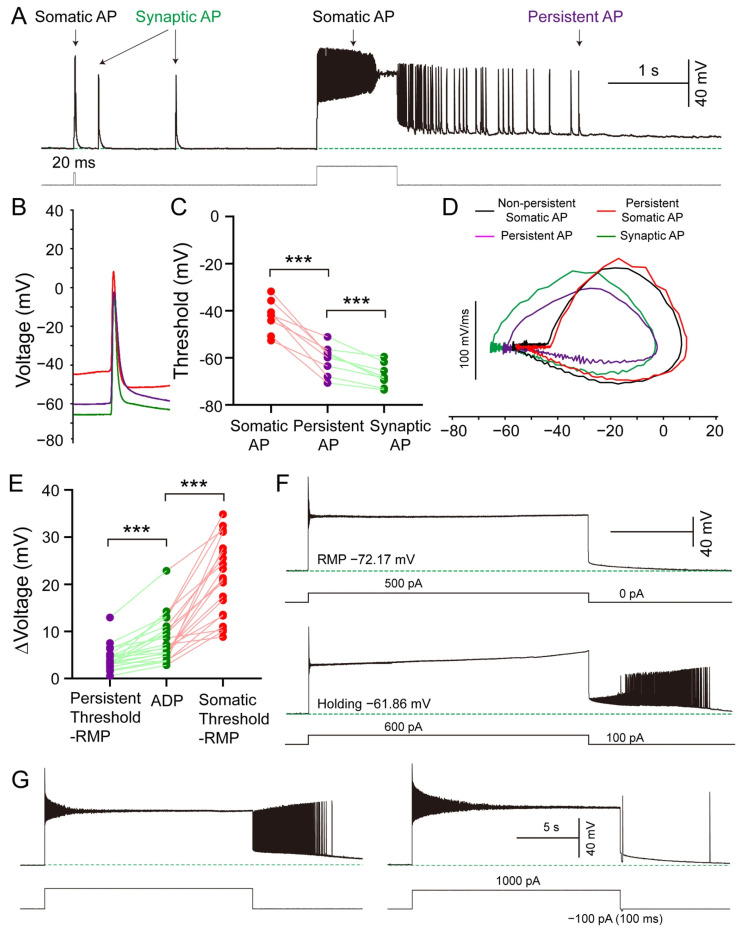
Persistent firing induced by L-ADP. (**A**) Typical traces of persistently firing neurons showing somatic AP (somatic APs refer to those directly evoked by depolarizing current injected through the recording electrode, the first AP), persistent AP (persistent APs refer to those occurring during the persistent firing period after the cessation of the evoked current) and synaptic AP (synaptic APs refer to spontaneous APs driven by synaptic inputs, the second and third AP). (**B**) Representative trace of somatic AP (red), persistent AP (magenta) and synaptic AP (green). (**C**) Quantification of AP threshold shown in (**B**). (**D**) Phase plot (dV dt^–1^ versus V) of APs from B. (**E**) Quantification of Δvoltage in persistent AP, L-ADP and somatic AP in individual persistently firing cells. (**F**) Typical traces of persistent firing induced by a holding potential of −60 mV. (**G**) Response to hyperpolarization stimulation during the persistent firing period (n = 3). ((**A**–**G**) n = 17 persistent firing neurons from 6 mice, *** *p* < 0.001 ).

**Figure 5 biomolecules-15-01603-f005:**
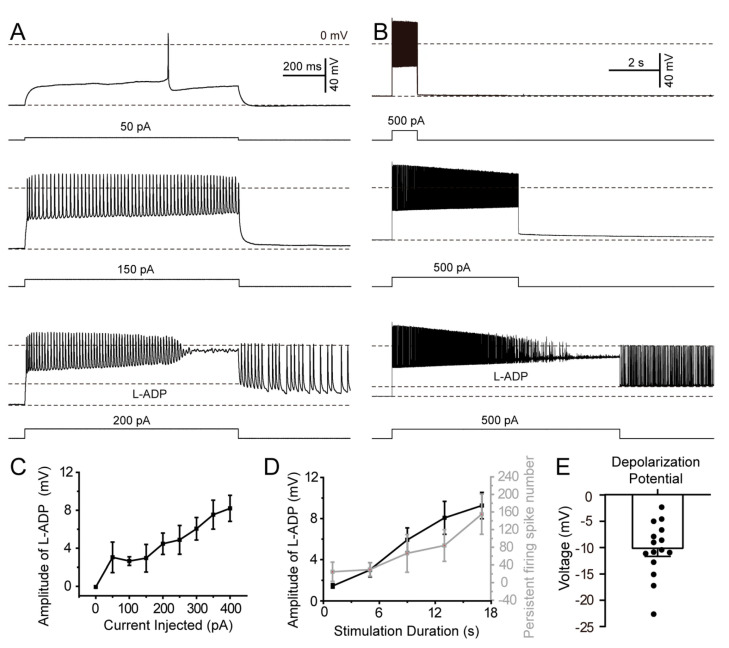
L-ADP amplitude regulated by stimulation intensity. (**A**) Typical traces showing L-ADP amplitude in response to depolarization currents; injected currents were increased from 0 to 400 pA in 50 pA steps. (**B**) Typical traces showing L-ADP amplitude in response to depolarization currents; stimulation duration ranged from 1 s to 13 s in 4 s step. (**C**) Relationship between injected current and amplitude of L-ADP. (**D**) Relationship of stimulation duration to amplitude of L-ADP and persistent-firing spike number. (**E**) Quantification of depolarization potential in individual persistently firing cells. (n = 15 persistent firing neurons from 6 mice ).

**Figure 6 biomolecules-15-01603-f006:**
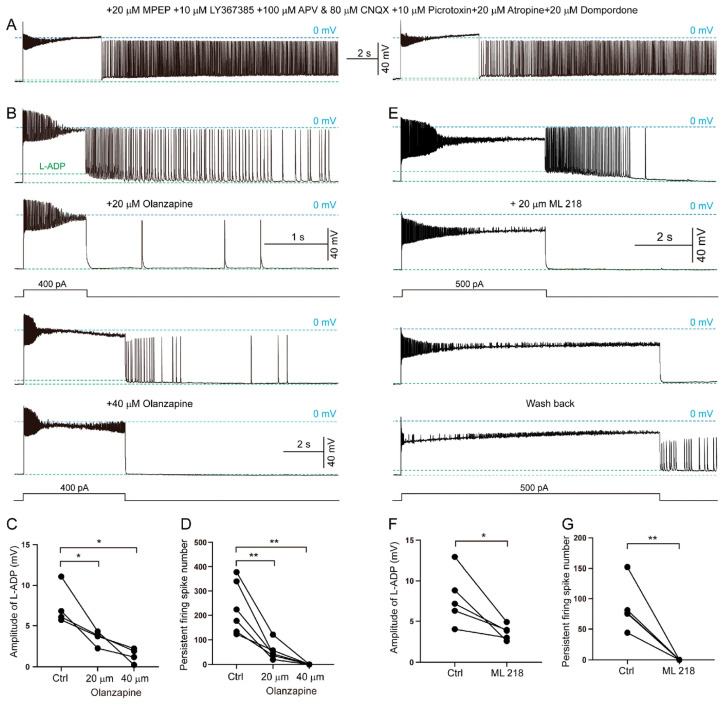
The synaptic and intrinsic cellular mechanisms of persistent firing in SLM neurons. (**A**) Inhibition of glutamatergic, GABAergic, cholinergic, dopaminergic receptors does not block persistent firing in SLM neurons. (**B**) Typical traces showing persistent firing inhibited by the 5-HT receptor inhibitor, olanzapine. (**C**) Typical traces showing persistent firing inhibited by the T-type calcium channel blocker, ML218. (**D**,**E**) Quantification of L-ADP amplitude and persistent-firing spike number following inhibition by olanzapine (n = 6, from 3 mice). (**F**,**G**) Quantification of L-ADP amplitude and persistent-firing spike number inhibited by ML218 (n = 5, from 3 mice, * *p* < 0.05, ** *p* < 0.01).

## Data Availability

The original contributions presented in this study are included in the article. Further inquiries can be directed to the corresponding author.

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
