# Peer review of "The Synaptic and Intrinsic Cellular Mechanisms of Persistent Firing in Neurogliaform Cells"

_biomolecules, 2025, doi:10.3390/biom15111603_

Round 1
Reviewer 1 Report
Comments and Suggestions for Authors
See doc

Reviewer 2 Report
Comments and Suggestions for Authors
In this article Chen and coworkers describe the results of their investigation about the cellular and network mechanisms underlying persistent firing in neurogliaform cells (NGFCs). Authors employing a combination of electrophysiological, pharmacological, and labeling techniques, conclude that the majority of NGFCs in the stratum lacunosum-moleculare display persistent firing, which is associated with long-lasting delayed afterdepolarizations, involves T-type calcium channels, and serotonin receptors. Persistent firing was also observed in a minority of PFC neurons.
Though the phenomenon of persistent firing has been previously investigated in principal neurons, we know relatively less in NGFCs which are GABAergic cells. In general, the manuscript present original data about persistent firing by linking this phenomenon with novel specific mechanisms, thereby adding information to the relevant current literature.
However, there is considerable room for improvement in this manuscript, as outlined below.
Specific Comments:
1). Given that the dorsal and ventral hippocampus differ in cellular physiology and neuromodulation, it should be specified that the slices were taken from the dorsal hippocampus, and it should also be noted that the slices were cut coronally.
2). The number of animals used in each experiment is reported by the authors, but not consistently. For example, this information should also be provided for all pharmacological experiments whose results are used in a critical way to support the conclusions.
3). Several results in this study are based on a particularly limited sample size, which should be interpreted with caution as it could affect the outcome of the statistical analyses. For example, they used only 5–6 cells in the pharmacological experiments investigating the involvement of L-type calcium channels and serotonin receptors. Therefore, the authors should either attempt to expand the sample size in the critical experiments or somewhat reduce the weight given to these results in the Discussion and Conclusions.
4). The involvement of network mechanisms proposed by the authors appears somewhat unclear and perhaps not well substantiated. Firstly, the authors base their suggestion of 5-HT receptors involvement on their results with olanzapine, an atypical antipsychotic. However, as they themselves note, this effect may not be specific. The use of a selective antagonist for these receptors would be necessary to demonstrate their involvement beyond doubt, since olanzapine also shows affinity for other receptors, such as dopamine, muscarinic, and several others. Moreover, an argument based solely on exclusion (i.e., the absence of effects via other receptors) is not sufficient to support the claim of serotonergic receptor involvement, which constitutes a key component of their conclusions. Also, if fast glutamatergic and GABAergic transmission are not involved in a phenomenon such as persistent firing, how could serotonergic receptors contribute to the “network” aspect of the phenomenon? Thus, the interpretation of a role for 5-HT–dependent network mechanisms should be made with caution, with a clearer acknowledgment of the limitations of the relevant experiments. In fact, this interpretation should probably not be used.
5). In addition, although the authors propose serotonergic receptors as a contributing mechanistic basis, the related literature references are rather outdated and relatively limited.
Reviewer 3 Report
Comments and Suggestions for Authors
The authors’ manuscript provides some new information about the causes of persistent firing of neuronal cells of the prefrontal cortex and of the hippocampal stratum lacunosum-moleculare. They demonstrate experimental hints for persisting firing being dependent on delayed afterdepolarization, T-type calcium channels and activation of 5-HT receptors. Complex slice voltage clamp measurements were necessary to obtain these experimental results. These are interesting findings which may have impacts on future therapies of CNS illness.
However, some details of the manuscript still need to be clarified before it can be published.
Major:
1.) P4 L127: “To check neuron morphology after electrophysiological recordings, in some cases, 0.2% biocytin (B4261, Sigma, USA) was added to the intracellular solution, allowing the visualization of patched cells. …”
How were the other cells characterized. Could the authors be sure that they really clamped neurogliaform cells?
2.) Fig. 1: What is the meaning of the green lines in A (right)? In C, it is probably the voltage without current injection.
Please give the current clamp protocol in all figures!
3.) Fig. 2C: Are the non-persistently firing cells NGFCs?
4.) Fig. 2E: The numbering is wrong. Figure labeled F does not exemplarily show the statement attributed to Figure D in the legend. The meaning of the shown example is not clear. The other recordings show current clamp with a constant current pulse.
5.) Fig. 3A: Why is there no firing after cessation of the current injection in the “Persistent” labeled neuronal recording? These are rather persisting somatic APs (see Fig. 4). Please clarify!
6.) P7 L219: Please explain somatic (induced by current injection) and synaptic (spontaneous) APs! What about the persistent AP threshold? The persistent APs occur without current injection! The difference between persistent somatic and the other persistent APs (after cessation of the current clamp?) should be made more clear.
7.) P9 L260: “Persistent firing was triggered by membrane depolarization to approximately −10 mV (range: −25 to −5 mV from resting potential; Figure 5E).”
Unclear, depolarization of -10 mV from RMP or to -10 mV?
8.) P11 L 328: “Both we and Sheffield et al. (2013) found that extracellular calcium may play a significant role in persistent firing, because EGTA or BAPTA in piped chelated the intracellular calcium and cannot block persistent firing.”
This is not a prove of the role of extracellular Ca2+!
Minor:
1.) P2 L58, “In contrast, within the GABAergic system, neurogliaform cells (NGFCs)— a specific subtype of interneurons expressing 5-hydroxytryptamine (5-HT)3a receptors— exhibit delayed persistent firing in response to depolarization stimuli, independent of exogenous neurotransmitters.”
What are the causes of the depolarization. Isn’t serotonin a neurotransmitter?
2.) P2 L61: Please explain retroaxonal barrage firing.
3.) P2, L66-72: Delete this paragraph. These facts have to be mentioned in the discussion and/or the abstract or summary.
4.) In most of the described results, the authors performed whole cell current clamp measurements, not patch clamp. Patch clamp would mean that a patch of the membrane would be voltage clamped. In Fig. 1A, left, whole cell voltage clamp measurements were shown. Please correct this.
5.) P6 L202: “However, persistent firing neurons exhibited broader APs (longer rise and decay times; Figure 3H, I), lower input resistance (Figure 3J), and reduced sag potentials (indicating diminished Ih currents; (Figure 3K).”
Could you explain this? At similar capacitance and lower input resistance, the membrane time constant should decrease. This would lead to shorter rise and decay times.
Please explain “sag potentials”.
5.) P7 L234: “a depolarization current was injected to clamp the membrane potential at -60 mV”
The potential is not really clamped. It is current clamp and you measure the voltage.
6.) Fig. 6: The labeling is hard to read.
7.) P1 L305: “Together, our data demonstrate for the first time that persistent firing in SLM NGFCs is supported by both network and intrinsic cellular mechanisms”
Please avoid “for the first time”. If it would not be for the first time it would not be worthy of publication in this journal.
8.) Please give the reference number for Sheffield et al.!
Round 2
Reviewer 1 Report
Comments and Suggestions for Authors
I think the work is interesting and itcan be published
Author Response
We thank Reviewer 1 for their positive feedback and endorsement of our work.
Reviewer 2 Report
Comments and Suggestions for Authors
The authors have addressed most of my concerns. However, I feel it is important to insist, from a scientific standpoint, on one key issue, specifically, the interpretation of the “network mechanism” of serotonergic action. As I pointed out previously, the involvement of 5-HT receptors does not constitute evidence for a true network mechanism, given that glutamatergic and GABAergic transmission were pharmacologically blocked. For this reason, I would suggest replacing the term “network mechanism” with the more scientifically accurate term “neuromodulatory mechanism”, “synaptic mechanism”, or a combination of these. This would preserve the precision and validity of the conclusions drawn from the presented results.
Author Response
We agree with Reviewer 2 that the term “network mechanism” may be misleading in the context of our pharmacological blockade. Accordingly, we have replaced “network mechanism” with “synaptic mechanism” throughout the manuscript to more accurately reflect the nature of our findings.
Reviewer 3 Report
Comments and Suggestions for Authors
see uploaded file

Round 3
Reviewer 2 Report
Comments and Suggestions for Authors
The authors have addressed my previous concern regarding the interpretation of the “network" mechanism of serotonergic action. The term has been appropriately revised to “synaptic" mechanism, which provides a more scientifically accurate and precise description consistent with the experimental conditions. I have no further concerns.
Author Response
We thank Reviewer 2 for their positive feedback and endorsement of our work.
Reviewer 3 Report
Comments and Suggestions for Authors
see file

Author Response
We thank the reviewer for the careful reading of our manuscript and the constructive comments. We have addressed each of the points raised as detailed below:
- Comment on Figure 1A:
“In Fig. 1A, the green line can not represent the resting potential, since the measurement was performed under voltage clamp. Are the authors sure what they are measuring?”
Response: We appreciate the reviewer’s astute observation. The green line in Figure 1A was intended to indicate the baseline membrane potential at the -60 mV holding potential, rather than the resting potential. We have revised the figure legend to clarify that the green line represents the baseline voltage level.
- Comment on Figure 2F and legend:
“The legend to Fig. 2F should be rewritten like: “F: Representative record of persistent firing at a frequency 10 Hz).” The statement “Persistent firing was induced by stimulation at frequencies ranging from 10 Hz to 100 Hz” should not be part of the legend, rather part of the Results text.”
Response: We agree with the reviewer’s suggestion. We have revised the legend of Figure 2F:
“F: Representative record of persistent firing induced by 10 Hz stimulation.”
The sentence describing the frequency range has been moved to the Results section as recommended.
- Comment on Figure 3A:
“In Fig. 3A, the time course of the current injection must be given. Furthermore, the meaning of the gray dashed lines has to be explained.”
Response: We thank the reviewer for this important suggestion. In response, we have now added the current injection protocol waveform directly to Figure 3A to clearly show the timing and duration of the stimulus.
Additionally, we have expanded the Figure 3 legend to explicitly state the meaning of the gray dashed lines in each context:
For the +50 pA trace, the gray dashed lines represent the measurement of the fast afterhyperpolarization (fAHP).
For the -150 pA trace, the gray dashed lines represent the measurement of the sag potential.
- Comment on Figure 4A:
“The characterization of the second AP as a synaptic one has to be clarified in the Figure.”
Response: We thank the reviewer for this suggestion. We have updated Figure 4A to explicitly label the second and third action potentials as “synaptic APs” in the figure itself, in addition to the revised legend which now clearly distinguishes between somatic and synaptic APs.
- Comment on membrane time constant and AP kinetics:
“This should be stated in the discussion.”
Response: We fully agree. We have now added the following paragraph to the Discussion section:
“Moreover, another finding of our study was that the kinetics of APs did not strictly follow predictions based on passive membrane properties alone. Specifically, we observed that a lower membrane time constant (tm) did not invariably result in shorter AP rise and decay times. While a reduced tm, resulting from a lower input resistance, should in principle accelerate the kinetics of passive membrane charging, the AP is an active response governed by voltage-gated ion channels. The rise time is primarily a function of voltage-gated sodium (NaV) channel activation kinetics, while the decay time is shaped by NaV channel inactivation and the activation of voltage-gated potassium (KV) channels. Therefore, our results strongly suggest that the differences in AP kinetics we report are dominated by cell-type or compartment-specific variations in the density, subtype composition, or regulatory states of these key ion channels. This highlights that active ionic mechanisms can override the modulating influence of passive cable properties in determining the temporal profile of neuronal spikes.”